# Determination of Modified QuEChERS Method for Chlorothalonil Analysis in Agricultural Products Using Gas Chromatography–Mass Spectrometry (GC-MS/MS)

**DOI:** 10.3390/foods12203793

**Published:** 2023-10-16

**Authors:** Da-Young Yun, Ji-Yeon Bae, Chan-Woong Park, Gui-Hyun Jang, Won-Jo Choe

**Affiliations:** 1Food Safety Evaluation Department, Pesticide and Veterinary Drug Residues Division, National Institute of Food and Drug Safety Evaluation, Ministry of Food and Drug Safety, Cheongju 28159, Republic of Korea; dyyun96@korea.kr (D.-Y.Y.); jiyeon0962@korea.kr (J.-Y.B.); arion@korea.kr (G.-H.J.); 2Center for Food and Drug Analysis, Busan Regional Office of Food and Drug Safety, Busan 47537, Republic of Korea; pcw0324@korea.kr

**Keywords:** chlorothalonil, monitoring, modified QuEChERS

## Abstract

Chlorothalonil is an organochlorine fungicide that blocks the respiratory process of cells and persists in agricultural products because it is used extensively to prevent fungal diseases. An analytical method of chlorothalonil using the modified QuEChERS method and gas chromatography– mass spectrometry (GC-MS/MS) was developed to analyze the residue in agricultural commodities distributed in Republic of Korea. Acetonitrile, including acetic acid and formic acid, was used to compare the extraction efficiency. The extraction and purification processes were established by comparing three versions of the QuEChERS method and various dispersive solid-phase extraction (d-SPE) combinations. Ultimately, 1% formic acid in acetonitrile with QuEChERS original salts and d-SPE (PSA, C_18_) were selected for the extraction and clean-up procedures for method validation and establishment. Five agricultural commodities, viz., brown rice, mandarin, soybean, pepper, and potato, were examined to validate the established method, which displayed excellent linearity, with a coefficient of determination of R^2^ = 0.9939–0.997 in the calibration curve range of 0.002–0.1 mg/kg. The limits of detection (LOD) and quantification (LOQ) were calculated to be 0.003 mg/kg and 0.01, respectively, for the method. The LOQ value satisfied the suitable level for the Positive List System (PLS). The mean recovery of chlorothalonil was 79.3–104.1%, and the coefficient of variation was <17.9% for intra- and inter-day precision at 0.01, 0.1, and 0.5 mg/kg. The matrix effects in the five commodities were confirmed by the ion suppression effects, except for brown rice, in which a medium enhancement effect was observed at 21.4%. Chlorothalonil was detected in eight apples, one watermelon, and one cucumber. Ultimately, chlorothalonil was detected in ten agricultural products. Thus, this analytical method could be used for the routine detection of chlorothalonil in agricultural products, and the data may be used to inform and improve current food policies.

## 1. Introduction

Fungicides are chemical compounds or biological organisms that can eliminate or hinder the growth of fungi and fungal spores [1]. Thus, these pesticides have been used to control plant diseases in agricultural systems for several decades. Fungicides have been reported to persist in food and environmental matrixes for prolonged durations because of their extensive application. Moreover, these residual components are toxic, and their toxicity is not restricted to targeting pests as it also has an impact on mammals, including humans [1]. Chlorothalonil is an organochlorine fungicide that blocks the respiratory process of cells primarily by inhibiting glycolysis via interactions with glutathione or glyceraldehyde 3-phosphate dehydrogenase. In particular, this pesticide is effective in preventing the germination of fungal spores and is used on vegetables and fruits, including tomatoes, celery, bananas, cucumbers, and beans [2]. In the United States, chlorothalonil was initially registered in 1966, and 384 tons are used annually for vegetable cultivation and on golf courses [2,3,4].

Chlorothalonil is a carcinogen that can affect the skin and intestine in animals and humans and can persist in soil for up to 100 days or 1 year because of its frequent use in crop cultivation. The residue, including the parent compound and its metabolites, can also be detected in fruits and vegetables [4].

In the Risk Assessment Report of the Joint FAO/WHO Meeting on Pesticide Residues (JMPR) for the evaluation of the toxicity of chlorothalonil, the acceptable daily intake (ADI) was estimated to be 0–0.02 mg/kg bw, and the acute reference dose (ARfD) was 0.6 mg/kg bw based on the no observed adverse effect level (NOAEL) from a two-year-long kidney toxicity study in rats [5]. The Codex Alimentarius Commission (CAC) defines only the parent compound as a residue of chlorothalonil and has an acceptable range of 0.01–70.0 mg/kg for 35 agricultural commodities [6]. The Japan Food Chemical Research Foundation (JFCRF) has established a maximum residue limit (MRLs) range of 0.01–25 mg/kg for 134 agricultural commodities [7], while the United States Environmental Protection Agency (EPA) has established an acceptable range of 0.05–15 mg/kg for 44 types of commodities. Additionally, the EPA has established MRLs for the 4-hydroxy-2,5,6-trichloroisophthalonitrile metabolite and chlorothalonil for hazelnuts, peppermint, persimmons, and spearmint [8].

In Republic of Korea, MRLs for chlorothalonil were established for 56 agricultural products, and the Positive List System (PLS) was applied for products that do not have established MRLs [9,10]. The PLS was implemented to enhance food safety regulations by uniformly applying a limit of 0.01 ppm to pesticides that do not have established MRLs in Korea. Initially the limit was established for nut seeds and tropical fruits in December 2016 and has been increasingly applied to all agricultural products since January 2019 [10].

The Ministry of Food and Drug Safety of Korea (MFDS) presently uses an analytical method to detect chlorothalonil in various matrixes, such as grains, potatoes, legumes, fruits, and vegetables, and the limit of quantitation (LOQ) was established at 0.01 mg/kg. The analysis involves the use conventional methods such as liquid–liquid extraction using acetone, hexane, dichloromethane, and NaCl solution, followed by purification using a column filled with Florisil^®^ and subsequent analysis via gas chromatography with an electron capture detector (GC-ECD) [11]. However, this procedure is time-consuming and costly because it requires large quantities of experimental material and does not guarantee a sufficient level of analysis [12].

A widely accepted analytical methodology for pesticide quantification is the QuEChERS (Quick, Easy, Cheap, Effective, Rugged and Safe) method [11]. The QuECHERS method is an analytical method published by Anastassiades in 2003 and consists of liquid–liquid partitioning extraction using acetonitrile and purification using dispersive solid-phase extraction (d-SPE). QuEChERS consists of two simple steps: liquid–liquid partitioning using acetonitrile and salts for extraction, followed by a clean-up phase using d-SPE. This technology is popularly used as a cost-effective and rapid multi-residue pesticide analysis technique in various sample matrixes, such as food and soil [12,13].

Several studies have used the QuECHERS method for chlorothalonil analysis; when utilizing extractive solvents with varying ratios of acetonitrile, toluene, and acetic acid, the recovery of chlorothalonil in cabbage was excellent, ranging from 71 to 93%. However, some studies have showed that vegetables such as tomato, garlic, and leek can be extracted using ethyl acetate via the QuEChERS method, and tomato showed excellent recovery (70–121%), while the others exhibited poor recovery (less than 61%); therefore, it is necessary to develop an adaptable technology by modifying the QuEChERS method to improve chlorothalonil analysis in food matrixes [14,15].

This study aims to develop an optimal analytical method for the quantification of chlorothalonil using acidified acetonitrile and the QuEChERS method (Original, AOAC, and EN) using GC-MS/MS. Representative agricultural products such as brown rice, soybeans, mandarins, potatoes, and peppers were used to validate the chlorothalonil method [16,17]. Furthermore, we intend to continuously monitor the safety of agricultural products distributed in Korea by utilizing the resulting data as scientific evidence to inform and improve food policies.

## 2. Materials and Methods

### 2.1. Chemicals and Reagents

Chlorothalonil (99.5% purity) was purchased from Sigma-Aldrich (Buchs, Switzerland), and its physicochemical characteristics are listed in Table 1. HPLC-grade acetonitrile was obtained from Merck (Darmstadt, Germany). Formic acid was obtained from Daejung (Siheung, Republic of Korea). QuEChERS salt extraction kits (Original, EN 15662, AOAC 2007.01) were purchased from Chromatific (Heidenrod, Germany). Primary secondary amine (PSA) was purchased from Agilent Technologies (Santa Clara, CA, USA), and octadecylsilane (C_18_) and graphitized carbon black (GCB) were purchased from Waters (Leinster, Ireland). The polytetrafluoroethylene syringe filters (PTFE, 0.45 µm pore size) were purchased from Thermo Fisher Scientific (Waltham, MA, USA).

### 2.2. Sample Selection and Collection

The agricultural commodities used to establish the analytical method were cereals (brown rice), legumes (soybean), potatoes (potatoes), fruits (mandarins), and vegetables (pepper), in accordance with Korean guidelines [16,17]. These five products were analyzed to confirm the presence of non-detected target pesticides and used for method validation. Agricultural products for the monitoring of chlorothalonil were collected from Seoul (47), Busan (43), Incheon (35), Daegu (31), Daejeon (24), Suwon (20), Ulsan (20), Gwangju (20), Cheongju (16), Changwon (18), Wonju (14), Jeju (12), Jeonju (10), and an online market (30) in Republic of Korea. A total of 15 commodities were selected for monitoring, and 340 samples were collected from local markets: vegetables (154), fruits (91), potatoes (47), cereals (34), and legumes (14). Those were homogenized following the procedure outlined in the Food Code and placed in a sealed container at −20 °C [11].

### 2.3. GC-MS/MS Analytical Conditions

Gas chromatography was performed for chlorothalonil analysis using an Agilent 7890 B GC system coupled to a 7010 B triple quadrupole (Agilent Technologies, Santa Clara, CA, USA). Agilent MassHunter QQQ Acquisition and Quantitative Analysis software version 10.0. was used for data acquisition and quantification. The column used for analysis was a DB-5MS UI (30 m × 0.25 mm, 0.25 µm, Agilent, Santa Clara, CA, USA). High-purity helium was used as a carrier gas at a flow rate of 1.5 mL/min. The injection mode was splitless, and the volume was 1 µL. The interface temperature was 280 °C. The conditions of the oven temperature are presented in Appendix A.

### 2.4. Sample Preparation

Each commodity was weighed to 10 g (with the exception of 5 g of both brown rice and soybeans) in a 50 mL centrifuge tube. Dry samples (brown rice and soybeans) were wetted with 10 mL of 1% formic acid in water for 30 min; thereafter, 10 mL of 1% formic acid in acetonitrile was added to each of the five samples. The mixture was vortexed for 1 min, added to the original salt extraction kit containing 4 g MgSO_4_ and 1 g NaCl, and shaken for 1 min. The sample was centrifuged at 4000× *g* for 10 min, and 1 mL of the supernatant was transferred to a d-SPE tube containing 150 mg MgSO_4_, 25 mg PSA, and 25 mg C_18_ to optimize the clean-up procedure. The tube was vortexed for 30 s and centrifuged at 4000× *g* for 10 min. The supernatant was filtered through a syringe filter (PTFE, 0.45 µm) and transferred to GC vials for GC-MS/MS analysis.

### 2.5. Method Validation

Chlorothalonil (10.03 mg) was dissolved in 10 mL of 1% formic acid in acetonitrile to prepare a stock solution. The stock solution was diluted with 1% formic acid in acetonitrile to make standard solutions for a calibration curve (0.02, 0.05, 0.1, 0.2, 0.5, 1.0 mg/L). The solutions were stored in the dark at −4 °C. To prepare the matrix-matched standards (0.002, 0.005, 0.01, 0.02, 0.05, 0.1 mg/L), the standard solutions were diluted by using a 90% blank sample which was a non-spiked extract.

Based on these guidelines, the selectivity, accuracy, precision, linearity, limit of detection (LOD), and LOQ were validated for the analytical method for chlorothalonil. For the selectivity, the presence of interfering substances was investigated by comparing the non-spiked and spiked samples. Accuracy can be determined by the results of the analytical recovery, and precision represents the degree of dispersion of the repeated results (relative standard deviation, RSD), which is expressed as a coefficient of variation (CV, %). The validation of the analytical method was performed by spiking agricultural products five times at concentrations corresponding to the LOQ, 10 × LOQ, and 50 × LOQ, and the accuracy and precision (intra- and inter-day precision) were confirmed by calculating the average recovery (%) and CV (%). To determine the linearity, a calibration curve at a concentration range of 0.002–0.1 mg/L was prepared using a matrix-matched method, and the coefficient of correlation (R^2^) and the linear equation of the calibration curve were calculated. The LOD and LOQ were obtained via the minimum detection concentration from the signal-to-noise ratios (*S*/*N* ratios) of 3 and 10, respectively [17,18]. In addition, the matrix effect was confirmed by comparing the slope of the standard curve of the sample with that of the solvent to confirm the suppression or enhancement effect of each sample. The formula is shown below [16]:$$ME\left(\%\right)=\left[\frac{Slope\;of\;spiked\;in\;matrix-matched\;calibration\;curve}{Slope\;of\;spiked\;in\;based-solvent\;calibration\;curve}-1\right]\;\times \;100.$$

## 3. Results and Discussion

### 3.1. Optimization of Analytical Conditions in GC-MS/MS

Chlorothalonil is volatile and thermally stable at ambient temperature and can be analyzed via GC. It has a structure containing four chlorine and two nitrile groups (Table 1) and is known to be difficult to detect in various sample matrixes; thus, several studies have analyzed chlorothalonil using an GC-ECD [11,14]. According to the EURL report, there is a study which analyzed chlorothalonil using LC-MS/MS, but there was no marked difference between using the atmospheric pressure chemical ionization (APCI) mode rather than the electrospray ionization (ESI) mode generally used for multi-residue pesticide analysis and GC-MS analysis [19,20,21,22]. In addition, GC-MS/MS was selected as an analytical device that can secure analysis sensitivity even at a relatively high selectivity and low concentration level and secure an LOQ of 0.01 mg/kg due to the introduction of the PLS. The analytical column selected was the DB-5MS UI (30 m × 0.25 mm, 0.25 μm), which is commonly used for multi-residue analysis due to its high selectivity and meets the 0.01 mg/kg level of LOQ [20,22]. Sample injection was performed in the splitless mode, and the ionization method for the target component was electron ionization (EI). The optimal characteristic ion for multiple reaction monitoring (MRM) was selected using full and product ion scans. The exact mass of chlorothalonil was 263.9 g/mol, and the optimal precursor ions were 266, 264 m/z; 266 m/z of product ion was 133, 170 m/z, and 264 m/z of product ion was 168 m/z. The selected characteristic ions confirmed the optimal collision energy (CE) required to establish the final analysis conditions listed in Table 2.

### 3.2. Optimization of Sample Preparation

#### 3.2.1. Selection of Extraction

Chlorothalonil was extracted with acetone using liquid–liquid partitioning and Florisil^®^ column chromatography in the conventional analytical method used for analysis in Republic of Korea [11]. However, this method has a long pretreatment time because it uses 25 g of samples and 100 mL of extraction solvent due to the easily volatile nature of acetone. Additionally this analytical procedure involved liquid–liquid partitioning, and dichloromethane and saturated NaCl solution were used, as well as florisil column chromatography for the cleaning up of samples; thus, it is a time-consuming and non-economical analysis method. Furthermore, the separation from the aqueous solution layer is lower than that when using highly polar solvents such as acetonitrile, which may reduce the recovery rate of pesticide components with medium or high polarity. By contrast, the QuEChERS method was compared with the conventional technique and thought to improve the method; the pretreatment time and reagent (acetonitrile) consumption were reduced. Additionally, acetonitrile extracted lower quantities of co-extracts and was easier to separate from the aqueous layer than other nonpolar solvents [23,24,25].

QuEChERS is economical and easily extracts samples <10 g with a low consumption of reagents [23,24]. In the QuEChERS method, acetonitrile, which is used as an extraction solvent, has limited solubility in lipids, and this could have been proven problematic in the recovery of pesticides from fatty commodities [23]. Thus, soybeans, which are classified as high oil or fat samples, were used to determine the extraction and purification conditions [16,17].

Based on the characteristics of the samples and pesticides, various QuEChERS methods (original, AOAC 2007.01, EN 15662) were used to compare extraction efficacies. Upon comparing the extraction efficacies of the three QuEChERS methods, we first considered the wetted samples during pretreatment. As shown in Figure 1A, all the soybean samples wetted with non-acidified water exhibited a low recovery (<20%). In the pesticide manual, chlorothalonil extraction was performed above pH 9 and the chlorothalonil was in the unstable state [26]. Therefore, 1% acetic or formic acid was added to the wetting and extraction solvents to maintain acidic conditions. Upon comparison of the three QuEChERS methods with different wetting and extraction solvents, the original method displayed significantly better recovery. The best combination was confirmed to be wetting combined with extraction solvent containing 1% formic acid in the original QuEChERS method (109.7% recovery) (Figure 1). Therefore, these extraction conditions were selected for subsequent clean-up experiments.

#### 3.2.2. Selection of Clean-Up Procedure

The clean-up procedure using dispersive solid-phase extraction (d-SPE) was reviewed to remove the impurities present in the extract. d-SPE involved the use of magnesium sulfate (MgSO_4_) and various types and amounts of adsorbents; d-SPE is a purification method suitable for these types of samples [23,25]. Comparing the recoveries obtained when using various combinations of PSA, octadecylsilane (C_18_), and GCB with MgSO_4_, the highest recovery was 102.7% in the refining method containing PSA and C_18_ (Table 3). These results suggest that C_18_, which effectively removes interfering substances such as lipids, and PSA, which removes organic acids and sugars [23], work best in combination in chlorothalonil purification. When GCB is used, some components with planar molecular structures are adsorbed and reduce the recovery rate; however, the planar structure of chlorothalonil is also adsorbed, but the recovery rate is lower than those of the other d-SPE combinations [12,14]. Based on the above results, a test method was established for extraction via the original method using acetonitrile containing 1% formic acid (with water containing 1% formic acid for brown rice and soybeans). The combination of adsorbents for d-SPE was refined to 150 mg of MgSO_4_, 25 mg of PSA, and 25 mg of C_18_.

### 3.3. Method Validation

#### 3.3.1. Selectivity, Linearity, Matrix Effect, Limit of Detection, Limit of Quantification

The interfering substance was not detected at the same retention time and mass-to-charge ratio (*m*/*z*) of chlorothalonil in the blank and spiked samples; therefore, the method was determined to have high separation and selectivity (Figure 2).

The stock solution was diluted with 1% formic acid in acetonitrile, and six concentrations of standard samples (0.002, 0.005, 0.01, 0.02, 0.05, 0.1 mg/L) were prepared to check the linearity and matrix effects of the five agricultural products. Standard samples (1 uL) were injected into the GC-MS/MS system, and an excellent coefficient of determination (R^2^) of >0.99 was confirmed for all of the agricultural products analyzed.

The ionization efficiency of the target analyte can be altered by the matrix, resulting in ion suppression or enhancement, and may affect the GC-MS/MS analytical method. The matrix effect is indicated as positive or negative and is typically classified as small (−20 < ME < 20), medium (−50 < ME < −20, 20 < ME < 50), or strong (ME < −50, ME > 50) [27,28]. Except for brown rice, all food matrixes were observed to be in the ~−21.5 to −51.4% range and indicated a signal suppression effect. Chlorothalonil displayed a strong suppression value in soybean samples at −51.4% (Table 4). Signal enhancement was confirmed for brown rice, and a value of 21.4% indicated a medium matrix effect.

The LOD and LOQ of chlorothalonil were determined as signal-to-noise ratios (*S*/*N*) above 3 and 10, respectively, and were calculated as 0.003 mg/kg and 0.01 mg/kg in five agricultural products. The calculated LOQ would be suitable for determining whether the products complied with the PLS of 0.01 mg/kg.

#### 3.3.2. Accuracy and Precision

Chlorothalonil was spiked at the LOQ, 10 × LOQ, and 50 × LOQ to perform recovery experiments on the five representative agricultural products and evaluate the accuracy and precision of the developed method. The mean intra-laboratory recovery was 72.0–112.4%, and the CV was < 12.5%. Cross-validation was performed three times by the Busan Regional Office of Food and Drug Safety, and the average recovery and CV were calculated from the inter- and intra-laboratory validation results (83.5–104.1% and <17.9%, respectively) in Table 5. The validation results satisfied the Codex guidelines (CAC/GL 40-1993) and the practical manual of the MFDS in Korea [17,18].

### 3.4. Application of Developed Method in Monitoring

The developed analytical method for chlorothalonil was used to monitor agricultural products and evaluate its suitability for routine examinations. A total of 340 commodities were collected from domestic markets in Republic of Korea (34 rice, 31 apples, 30 onions, 29 radishes, 28 potatoes, 26 mandarins, 24 tomatoes, 22 cucumbers, 19 sweet potatoes, 18 persimmons, 17 green onions, 16 Korean melons, 16 watermelons, 16 bananas, and 14 soybeans). All commodities were applied in MRLs, and 0.01 mg/kg was used as the PLS if the MRLs were not established for the crops. Chlorothalonil was detected in ten crops (Table 6): eight apples, one watermelon, and one cucumber. Apples have a high detection rate (80%), and some studies have demonstrated that chlorothalonil can be maintained for up to a month after final treatment on the surface of the apples [28]. Moreover, chlorothalonil is known to be used in post-harvest treatment as a chemical control agent [29]; thus, it is expected that it can be used to yield high detection rate in apples. However, the results of one study indicated that residual chlorothalonil was significantly removed from the apple surface after washing [30]. Watermelons and cucumbers are classified as cucurbits, and chlorothalonil is used to protect against gummy stem blight, powdery mildew, and anthracnose infections [31]. Given the low detection rate compared with the number of collections and absence of agricultural products exceeding MRLs, it is predicted that the risk of chlorothalonil may be lower upon undergoing processes such as washing and cooking.

## 4. Conclusions

In summary, an analytical method for chlorothalonil was developed for agricultural product application using a modified QuEChERS method and gas chromatography–mass spectrometry (GC-MS/MS). Acetic and formic acids were used to compare the extraction efficiency of the wet and extraction solutions for pH adjustment, with 1% formic acid in water and 1% formic acid in acetonitrile being the most efficient extraction solutions. The original QuEChERS method extracted the pesticide most effectively, and the d-SPE combination, which comprised a combination of PSA and C_18_, showed the highest recovery rate for chlorothalonil. We validated the method in five crops (brown rice, mandarin, soybean, pepper, potato); excellent linearity (R^2^ ≤ 0.99) was accomplished, and the LOD and LOQ satisfied the PLS level with values of 0.003 mg/kg and 0.01 mg/kg, respectively. The mean recovery of chlorothalonil was 79.3–104.1%, and the CV was below 17.9% for intra- and inter-day precision at the LOQ, 10 × LOQ, and 50 × LOQ concentration levels. The validation results of the established method confirmed that the selectivity, accuracy, precision, LOD, and LOQ satisfied the guidelines. Furthermore, when the developed method was applied to distributed agricultural products, chlorothalonil was detected in ten agricultural products; therefore, it is expected that this analysis method can be used for the routine inspection of agricultural products.

## Figures and Tables

**Figure 1 foods-12-03793-f001:**
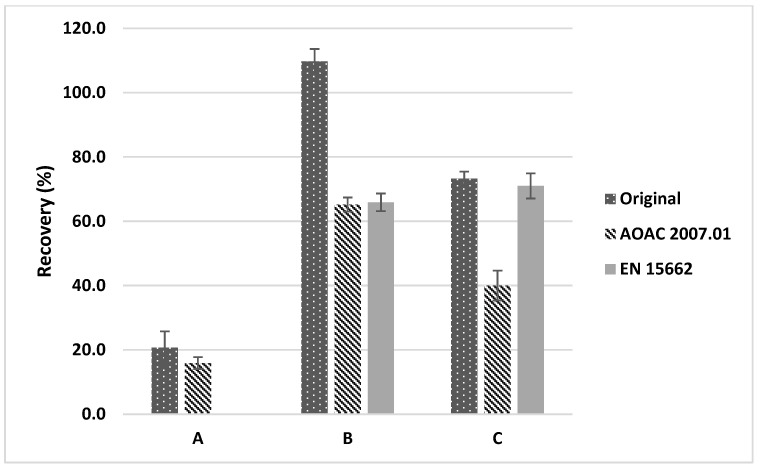
Comparison of extraction efficiency in soybeans for the determination of chlorothalonil by using different QuEChERS salts and extraction solutions: (**A**) water (wet solution) and acetonitrile (extraction solution); (**B**) 1% formic acid in water (wet solution) and 1% formic acid in acetonitrile (extraction solution); (**C**) 1% acetic acid in water (wet solution) and 1% acetic acid in acetonitrile (extraction solution).

**Figure 2 foods-12-03793-f002:**
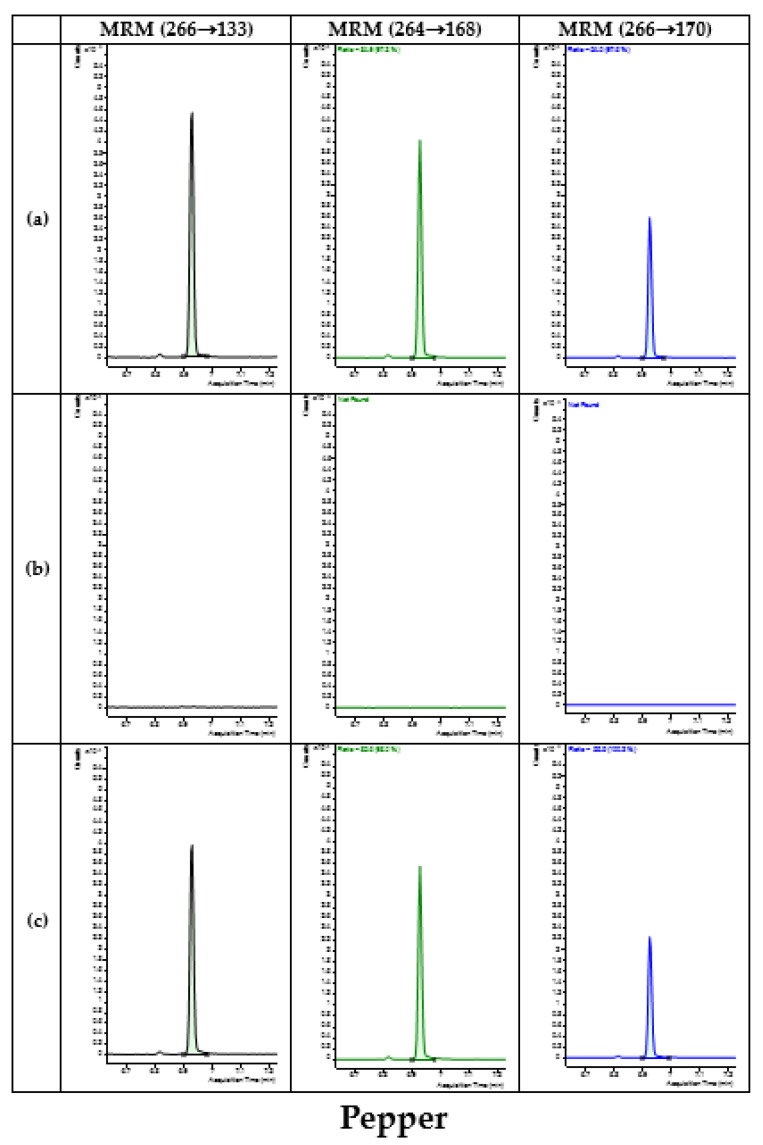
Representative MRM (quantification ion 266 > 133, qualification ion 264 > 168 and 266 > 170) chromatograms of chlorothalonil in five matrixes (pepper, mandarin, potato, brown rice, and soybean). (**a**) Matrix-matched standard at 0.01 mg/kg; (**b**) blank sample (**c**) spiked samples at 0.01 mg/kg.

**Table 1 foods-12-03793-t001:** Physicochemical characteristics of chlorothalonil.

Analyte	CAS No.	Molecular Formula	Structure	Vapor Point (mPa)
Chlorothalonil	1897-45-6	C_8_Cl_4_N_2_	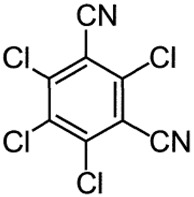	0.076

**Table 2 foods-12-03793-t002:** GC-MS/MS parameters for chlorothalonil.

Analyte	Molecular Weight (g/mol)	Exact Mass (g/mol)	Precursor Ion (*m*/*z*)	Product Ion (*m*/*z*)	Collision Energy (eV)
Chlorothalonil	265.9	263.9	266	133 ^(a)^	45
				170	30
			264	168	30

^(a)^ Quantification ion.

**Table 3 foods-12-03793-t003:** Recovery with different sorbent types for the determination of the optimal combination for chlorothalonil in soybeans.

Sorbent Type	Recovery (%)	CV ^a^ (%)
MgSO_4_ 150 mg, PSA 25 mg	93.3	10.4
MgSO_4_ 150 mg, C_18_ 25 mg	83.2	10.8
MgSO_4_ 150 mg, GCB 2.5 mg	58.2	9.0
MgSO_4_ 150 mg, PSA 25 mg, C_18_ 25 mg	102.7	5.1

^a^ Average coefficient of variation.

**Table 4 foods-12-03793-t004:** The analytical method of matrix-matched calibration and solvent calibration.

Matrix	Equation	R^2^	LOD	LOQ	Matrix Effect (%)
Solvent	y = 8622.5x + 6773	0.9968	0.003	0.01	-
Brown rice	y = 10,469x – 12,291	0.9959			21.4
Soybean	y = 4192.4x + 8532.3	0.9939			−51.4
Pepper	y = 6765.7x − 3063.6	0.9961			−36.0
Potato	y = 5795.2x + 3682.6	0.9971			−21.5
Mandarin	y = 5317.3x + 12,987	0.9964			−32.8

**Table 5 foods-12-03793-t005:** Validation results of the analytical method for chlorothalonil in food samples.

Matrix	Fortification (mg/kg)	Intra-Day		Inter-Day		Ave. ^a^	
		Recovery (%)	CV ^b^ (%)	Recovery (%)	CV ^b^ (%)	Recovery (%)	CV ^b^ (%)
Brown rice	0.01	89.6	7.0	101.1	4.1	95.4	8.5
	0.1	100.8	6.0	91.5	1.7	96.2	6.8
	0.5	109.2	5.8	89.3	2.6	99.3	14.2
Soybean	0.01	96.8	11.6	79.6	17.7	88.2	13.8
	0.1	72.0	3.1	86.5	13.0	79.3	12.9
	0.5	81.7	5.5	92.7	7.3	87.2	8.9
Mandarin	0.01	112.4	5.7	87.2	12.5	99.8	17.9
	0.1	96.5	11.6	93.9	2.8	95.2	1.9
	0.5	87.9	7.7	96.8	11.6	92.4	6.8
Potato	0.01	93.7	12.5	87.9	8.6	90.8	4.5
	0.1	89.3	8.3	99.3	12.2	94.3	7.5
	0.5	77.5	12.1	89.6	7.5	83.5	10.2
Pepper	0.01	89.0	1.3	111.9	6.2	100.5	16.1
	0.1	93.7	7.4	114.4	3.9	104.1	14.1
	0.5	84.7	7.5	94.2	3.6	89.5	7.5

^a^ Average recovery and coefficient values for 5 (intra-day) and 3 (inter-day). ^b^ Average coefficient of variation.

**Table 6 foods-12-03793-t006:** Monitoring chlorothalonil results for 340 marketed agricultural products in Republic of Korea.

Type	Commodity	No. of Samples	No. of Detected Samples	Concentration (mg/kg)
Fruits	Apple	31	8	0.01–0.10
	Banana	16	-	-
	Mandarin	26	-	-
	Persimmon	18	-	-
Vegetables	Cucumber	22	1	0.04
	Green onion	17	-	-
	Onion	30	-	-
	Radish	29	-	-
	Tomato	24	-	-
	Watermelon	16	1	0.08
	Korean melon	16	-	-
Cereals	Rice	34	-	-
Potatoes	Potato	28	-	-
	Sweet potato	19	-	-
Legumes	Soybean	14	-	

## Data Availability

The data presented in this study are available upon request from the corresponding author.

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
