# Peer review of "Determination of Modified QuEChERS Method for Chlorothalonil Analysis in Agricultural Products Using Gas Chromatography–Mass Spectrometry (GC-MS/MS)"

_foods, 2023, doi:10.3390/foods12203793_

Round 1
Reviewer 1 Report
The work of Yun et al., aimed to develop an analytical method for the detection of chlorothalonil in agricultural products distributed in South Korea using the modified QuEChERS method and gas chromatography-tandem mass spectrometry (GC-MS/MS) . The work was well done. The methods used were suitable and adequately described taking into account the limits of detection and quantification.
The results are interesting and the conclusions briefly summarize the data obtained and the aims of the work.
Author Response
I appreciate your valuable comments. Thank you.
Reviewer 2 Report
The manuscript titled "Determination of Modified QuEChERS Method for Chlorothalonil Analysis in Agricultural Products using GC-MS/MS" presented an analytical approach for the detection and quantification of chlorothalonil residues in agricultural commodities disseminated within the South Korean region. The modified QuEChERS method, in combination with gas chromatography-tandem mass spectrometry (GC-MS/MS), is employed for the analysis of these residues. The presence of matrix effects in the five commodities was verified through the observation of ion suppression effects, with the exception of brown rice, where a moderate enhancement impact was detected. The present approach possesses the potential to be employed for the regular identification of chlorothalonil in agricultural commodities. Prior to evaluating its appropriateness for publication in the journal "Foods," it is imperative to resolve the following considerations.
1. Could you please give a good reference for lines 79-81?
2. The authors have mentioned that “However, some studies have reported poor recoveries (52–70%) in tomato, leek, and garlic samples; therefore, it is necessary to develop an adaptable technology by modifying the QuEChERS method to improve chlorothalonil analysis in food matrixes [14,15].”. However, the specific optimisation results for tomato, leek, and garlic samples have not been provided. It is recommended to provide further clarification on this matter.
3. In lines 243-244, the author mentioned that “The combination of adsorbents for d-SPE was refined to 150 mg of MgSO4, 25 mg of PSA, and 25 mg of C18.” However, the main text does not give any information regarding the optimisation of the amounts of MgSO4, PSA, and C18. Therefore, it is necessary to include details regarding the optimisation of these amounts.
4. There is a lack of additional information regarding the detection method of Chlorothalonil, apart from the GC-MS/MS method. It is requested to provide a more detailed overview of the prior detection methods.
5. In the section titled "Optimization of Analytical Conditions in GC-MS/MS", the provided information lacks specificity about how it was employed to get the parameters. It is requested to provide more precise information in this regard.
6. In section 3.2.1, it was indicated that “the pretreatment time and reagent (acetonitrile) consumption were reduced.”. However, the specific information regarding the validation of these two points was not supplied. It is necessary to include further details regarding the validation process in order to enhance the comprehensiveness and rigour of the study.
7. The utilisation of magnesium sulphate (MgSO4) was employed in order to eliminate water from the extract and enhance the effectiveness of the extraction process. However, the examination of the effect of different sorbent materials in section 3.2.2 was limited to the usage of soybean. The matrix consisting of fruits and vegetables with a high water content has not been validated for optimised settings. Could you perhaps provide an explanation for this?
8. Where is the Table S1, as mentioned in section 2.3?
9. Please present a comprehensive workflow detailing the extraction, cleaning, and detection procedures employed in your method. This will enable readers to assess whether your approach is a superior alternative to established methods.
Author Response
1. Could you please give a good reference for lines 79-81?
I find that reference 12 provides valuable insights. Reference 12 elucidates, 'Traditional procedures are time-consuming, labor-intensive, complicated, and expensive; moreover, they produce considerable quantities of waste, and frequently, a sufficiently low limit of detection is unobtainable.' I would like to incorporate this quotation into lines 79-81.’ I upload to revise manuscript.
2. The authors have mentioned that “However, some studies have reported poor recoveries (52–70%) in tomato, leek, and garlic samples; therefore, it is necessary to develop an adaptable technology by modifying the QuEChERS method to improve chlorothalonil analysis in food matrixes [14,15].”. However, the specific optimisation results for tomato, leek, and garlic samples have not been provided. It is recommended to provide further clarification on this matter.
Yes, I will revise the sentence to incorporate your opinion.
However, some studies have performed that vegetables such as tomato, garlic and leek be extracted using ethyl acetate with the QuEChERS method and tomatoes showed excellent recovery (70-121%), while the others exhibited poor recovery, less than 61%; therefore, it is necessary to develop an adaptable technology by modifying the QuEChERS method to improve chlorothalonil analysis in food matrixes [14,15].
3. In lines 243-244, the author mentioned that “The combination of adsorbents for d-SPE was refined to 150 mg of MgSO4, 25 mg of PSA, and 25 mg of C18.” However, the main text does not give any information regarding the optimisation of the amounts of MgSO4, PSA, and C18. Therefore, it is necessary to include details regarding the optimisation of these amounts.
In lines 233-235, I discussed the comparison of recoveries achieved through different combinations of PSA, octadecylsilane (C18), and GCB, in conjunction with MgSO4. Among these combinations, the highest recovery rate observed was 102.7% in the refining method that included PSA and C18. Furthermore, taking into account the results presented in Table 3, it becomes evident that the combination of MgSO4, PSA, and C18 yielded the best results.
4. There is a lack of additional information regarding the detection method of Chlorothalonil, apart from the GC-MS/MS method. It is requested to provide a more detailed overview of the prior detection methods.
I explained prior detection methods of chlorothalonil in line 181-185. “According to the EURL report, there is a study which analyzed for chlorothalonil using LC-MS/MS, but there was no marked difference between using the atmospheric pressure chemical ionization (APCI) mode rather than the electrospray ionization (ESI) mode generally used for multi-residue pesticide analysis, and GC-MS analysis [20-22].”
5. In the section titled "Optimization of Analytical Conditions in GC-MS/MS", the provided information lacks specificity about how it was employed to get the parameters. It is requested to provide more precise information in this regard.
I submitted supplementary data along with the manuscript, which included Table S1, S2, and Figure S1 as part of the supplementary materials. I intend to upload the supplementary data once again, and it should be noted that Table S1 contains additional information pertaining to flow rate, injection mode and volume, and oven temperature program.
6. In section 3.2.1, it was indicated that “the pretreatment time and reagent (acetonitrile) consumption were reduced.”. However, the specific information regarding the validation of these two points was not supplied. It is necessary to include further details regarding the validation process in order to enhance the comprehensiveness and rigour of the study.
I intend to enhance the sentence below by providing a more detailed explanation.
However, this method take a long pretreatment time because that uses 25 g of samples and 100 mL of extraction solvent due to easily volatile property of acetone. Additionally this analytical procedure involved liquid-liquid partitioning which dichloromethane and saturated NaCl solution were used and the process of florisil column chromatography for sample’s clean-up, thus it is a time-consuming and non-economical analysis method.
7. The utilisation of magnesium sulphate (MgSO4) was employed in order to eliminate water from the extract and enhance the effectiveness of the extraction process. However, the examination of the effect of different sorbent materials in section 3.2.2 was limited to the usage of soybean. The matrix consisting of fruits and vegetables with a high water content has not been validated for optimised settings. Could you perhaps provide an explanation for this?
According to Reference 23, the QuEChERS method was originally introduced for use in pesticide analysis among high-water samples such as vegetables and fruits. However, because the solubility of lipids in acetonitrile, an extraction solvent of the QuECHERS method, is limited, experiments were conducted to optimize the extraction and purification of soybeans, a representative agricultural product of lipids. Fruits and vegetables were not subjected to optimization experiments because the recovery rate was viewed through the test method validation.
8. Where is the Table S1, as mentioned in section 2.3?
I submitted supplementary data along with the manuscript, which included Table S1, S2, and Figure S1, I will upload the supplementary data once again
9. Please present a comprehensive workflow detailing the extraction, cleaning, and detection procedures employed in your method. This will enable readers to assess whether your approach is a superior alternative to established methods.
I submitted supplementary data along with the manuscript, which included Table S1, S2, and Figure S1 as part of the supplementary materials. I intend to upload the supplementary data once again, and it should be noted that Figure S1 contains total workflow.

Reviewer 3 Report
During the review of the article entitled "Determination of Modified QuEChERS Method for Chloro- 2 thalonil Analysis in Agricultural Products using GC-MS/MS" highlighted the following: The structure of this, is well written, can be very useful to use this method, in the detection of the pesticide Chloro-2 thalonil, as well as others. The introduction is well described, but there is a conceptual error in line 85 where the name of the acronym (d-SPE) should be, which is only mentioned in line 89. The materials and methods are well cited and explained, each of them has method validation, but in my opinion the statistical methods using replicas are lacking. In Figure 1, the differences of the methods with the standard deviation are shown.
Author Response
Q : During the review of the article entitled "Determination of Modified QuEChERS Method for Chloro- thalonil Analysis in Agricultural Products using GC-MS/MS" highlighted the following: The structure of this, is well written, can be very useful to use this method, in the detection of the pesticide Chloro-2 thalonil, as well as others. The introduction is well described, but there is a conceptual error in line 85 where the name of the acronym (d-SPE) should be, which is only mentioned in line 89. The materials and methods are well cited and explained, each of them has method validation, but in my opinion the statistical methods using replicas are lacking. In Figure 1, the differences of the methods with the standard deviation are shown.
A : I have made revisions to lines 85 and 89 as requested. Additionally, I would like to highlight that in Table S2, we have provided data on recoveries along with their corresponding standard deviations. These results serve as the basis for the findings presented in Figure 1. I upload revising manuscript with supplementary data.

Round 2
Reviewer 2 Report
I can recomend its publication on this journal after checking the updated version with highlight modification.